# PRISM: A Paradigm for Controllable 3D Generation Driven by Structural Concept Prior

## Abstract

The generation of high-quality 3D assets is essential for applications in virtual reality, robotics, and industrial systems. Existing methods can be sorted into three categories based on different prior. The first lines lift 2D diffusion as prior into 3D representations. The second lines adopt ground truth multi-view images as prior to directly regress 3D assets. The third lines tend to model the probabilistic distribution of 3D assets, which adopt 3D distribution as their prior. However, those three types of prior are in semantic level. They can represent semantic information but ignore the structural concept (describing the topological structures), which is crucial in the physical world applications. To address this limitation, we propose a novel 3D generation paradigm, called Prism (a paradigm driven by structural concept), which leverages structural concept as prior. First, our method encodes structural concept which is fused with real-world images to form prior representations, enabling the model to integrate high-level structural concept prior while guaranteeing shape details from real-world images. Then we adopt a pre-trained VAE encoder to provide embeddings of real 3D models. After that, we employ consistency loss in the latent space to align our prior with real 3D models to achieve mapping between concept space and 3D space, ensuring the generated 3D assets are structurally coherent, aligned with affordance, and visually realistic. Prism provides a high shape quality and structure controllable solution for 3D synthesis. We validate our method on both vision and robotics aspects with state-of-the-art algorithms. Our code will be public available.

## 1 Introduction

The emergence of research such as robotics, virtual and augmented reality has increasingly brought the 3D assets generation into the spotlight of academic and industrial research. However, manual 3D assets creation is time-consuming, costly, and offers limited reusability. Consequently, the generation of 3D assets, often from a single image or a textual prompt, has become an active research field.

Existing 3D assets generation methods can be categorized into three paradigms. The first Poole et al. (2023); Lin et al. (2023); Tang et al. (2024) employs pre-trained 2D diffusion models with Score Distillation Sampling (SDS) to optimize Neural Radiance Fields(NeRF) Mildenhall et al. (2021). The second chooses to directly regress assets from sparse-view images Chan et al. (2022); Hong et al. (2024). The third is native 3D generation Li et al. (2025); Chen et al. (2025a); Xiang et al. (2025), which models the distribution of 3D assets directly, conditioned on text or images. Although these methods can produce visually realistic results, they lack structural concept prior, which can reflect the topological structures. As illustrated in 1, these methods are able to generate assets with good appearance, but they are difficult to highly control the topological structures.

We argue that a complete 3D asset should simultaneously capture two essential dimensions: fine-grained appearance and structural concept. Fine-grained appearance ensures realism and visual plausibility, while structural concept ensures controllable topological structure and functional usability, especially in downstream tasks involving interaction, such as robotic manipulation. Previous methods mainly focus on the appearance and do not pay much attention to the structural concept.

From a cognitive perspective Biederman (1987), objects can be described not only by their external appearance but also through basic geometry shape and topological structure. Inspired by this idea,

Figure 1: Comparison with our methods and previous 3D generation routes. (a)Previous types of prior are difficult in topological structures controlling. (b)Comparison between and previous methods, our method can control both shape details and topological structures (three evenly sized posts on the back of the chair), along with Affordance Knowledge described in procedures.

Analytic Concept Sun et al. (2024b) has been introduced to represent the structural concept by procedures, including topological structures and affordance knowledge which is illustrated in Figure 2 (details is described in Appendix A.3).

To this end, we propose a novel generation paradigm named Prism (a 3D generation paradigm driven by structural concept prior) that integrates structural concept prior based on Analytic Concept and shape detail prior based on real-world images, enabling the synthesis of 3D assets that are structurally coherent, visually realistic, and aligned in affordance knowledge with Analytic Concept. This paradigm faces three challenges: First, how to form prior representations based on Analytic Concept, which is novel in 3D generation. Second, how to control the topological structures of generated 3D assets while maintaining the shape details. Third, how to learn the mapping between the novel prior representations and 3D space.

To tackle these issues, we propose three core components: Prior Representation Construction, Shape Encoding, and 3D Space Mapping. For Prior Representation Construction, we construct prior representations based on Analytic Concept along with real-world images to control both topological structures and shape details. Specifically, the prior representations consists of vision features and language features integrated by cross attention, where the vision features is a fusion of conceptual image features and real-world image features via AdaLN Perez et al. (2018) modulation and the language features is extracted from the language descriptions of Analytic Concept.

For the Shape Encoding, a pre-trained VAE encoder is employed to encode the geometry of real 3D models into latent 3D space. Specifically, the geometry is represented by coarse shape obtained through uniform points sampling and geometric details captured via sharp points sampling, which are subsequently integrated through cross-attention to derive latent features.

For the 3D Space Mapping, we employ the 3D shape latents produced in the Shape Encoding process as supervision to learn a mapping between prior representations and 3D space. It should be noted that learning such a mapping process is essential because previous methods did not employ structural concept as their prior, which means that they cannot achieve such a mapping from prior representations to 3D space.

Through these three core components, Prism can generate 3D models that not only achieve high visual fidelity but also preserve structural concept prior, breaking through the limitations of previous approaches that mainly focused on semantic information. In summary, Prism establishes a novel paradigm that generates 3D models with both visual fidelity and structural concept, setting a solid foundation for applicability in both visual understanding tasks and real-world interactive scenarios. We believe this work opens a new avenue for controllable topological structures generation in the 3D domain.

Our work makes the following key contributions: (1) We propose a novel structural concept driven 3D generation paradigm, leveraging Analytic Concept as topological structures controllable prior. To the best of our knowledge, we are the first to employ structural concept as prior in 3D assets generation, ensuring structural controllable and potential affordance knowledge of the outputs. (2) To achieve the mapping between concept space and 3D space, we construct a novel prior represen-

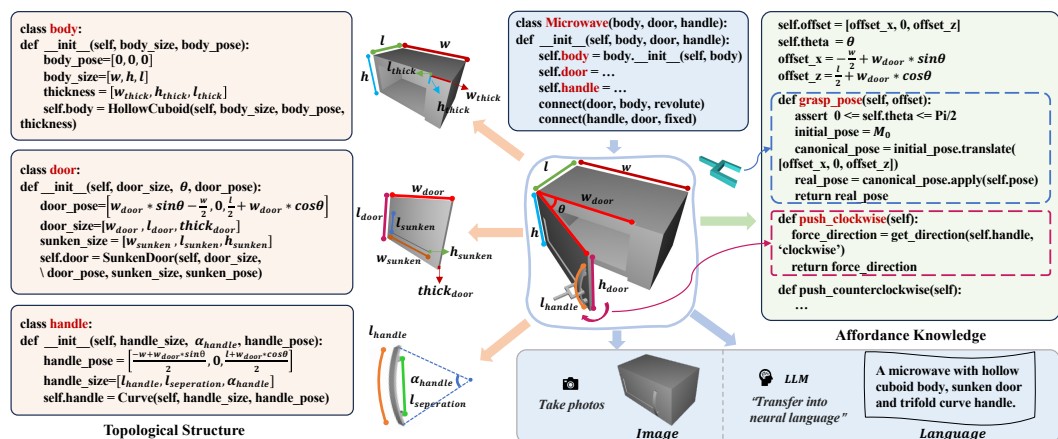

Figure 2: Illustration of Analytic Concept, which depicts topological structures and affordance knowledge in procedures. in procedures language. And can render into corresponding 3D model, render into image by camera parameters, or transfer into nature language description.

tations which has a one-to-one correspondence with the concept space, and then train a DiT to learn the mapping from prior representations to 3D space supervised by real 3D models. (3) To control both topological structures and shape details, we propose a multi-modal prior representation which consists of both vision and language features. And we employ AdaLN modulation and cross attention to merge the multi-modal features as our final prior representations. (4) We provide qualitative and quantitative evaluations to validate the benefits of Prism on both 3D generation and robotic aspects. The results demonstrate that our method can control both shape details and topological structures, while possess affordance knowledge described in Analytic Concept.

## 2 RELATED WORK

### 2.1 3D ASSETS GENERATION

3D assets generation methods are typically categorized into three paradigms: SDS-based optimization, sparse-view reconstruction, and native 3D generative models.

The first paradigm score-based optimization methods iteratively refine implicit 3D representations, most often instantiated as Neural Radiance Fields (NeRFs) Mildenhall et al. (2021), using supervision from pre-trained vision-language or diffusion models. DreamFusion Poole et al. (2023) and Magic3D Lin et al. (2023) apply Score Distillation Sampling (SDS) with 2D diffusion prior to optimize NeRFs. They improve visual fidelity and multi-view consistency but are slow and sometimes produce geometric artifacts . The second paradigm sparse-view reconstruction models reconstruct 3D assets from a small number of images Siddiqui et al. (2024); Xie et al. (2024); Wang et al. (2023). Instant3D Li et al. (2024) uses a transformer-based reconstructor to reconstruct NeRF from a small set of structural 2D views, enabling real-time generation with comparable quality. MVDream Shi et al. (2024) employs a multi-view diffusion model to generate coherent views from a single prompt, enhancing consistency when combined with score distillation sampling. The last native 3D generative models paradigm directly learn conditional mappings from text or images to 3D shape representations using paired data Zheng et al. (2023); Chen et al. (2025b). For instance, CLIP-Forge Sanghi et al. (2022) adopts an invertible normalizing flow to transform CLIP embeddings into shape embeddings. Dora Chen et al. (2025a) enhances reconstruction by prioritizing sharp-edge regions and employs dual cross-attention for fine detail preservation in autoencoding pipelines. Trellis Xiang et al. (2025), trained on a large dataset of diverse objects, yields a generalizable 3D generation model capable of producing multiple mainstream 3D formats for various applications.

Despite the remarkable performance of these 3D generation methods, they focus mainly on object's geometry with semantic prior, which leads to insufficient control ability of topological structures during generation.

## 2.2 MANIPULATION TASK

Object manipulation encompasses a range of tasks centered on enabling embodied agents to interact appropriately with objects. Pick-and-place is a fundamental task in manipulation. Some methods concentrate on pose estimation of known objects, while others choose to create universal policies for novel objects. Besides this, some researchers focus on improving the intelligence of robots to handle more complex tasks, such as manipulating deformable and articulated objects. For instance, Where2Act Mo et al. (2021) predicts per-pixel action likelihoods and manipulation proposals to guide interaction. Where2Explore Ning et al. (2023) adopts few-shot learning for object manipulation, which measures affordance similarity across object categories to transfer affordance knowledge to novel objects. Additionally, GAPartNet Geng et al. (2023) with semantic and affordance labels has been released, accompanied by a manipulation pipeline that leverages the concept of actionable parts. The effectiveness of manipulation using these methods is largely contingent on the depth of understanding regarding affordances in objects.

## 3 METHOD

In Section 3.1, we will present an overview of our method. In Section 3.2, we will elaborate on the proposed structural-concept-prior-driven paradigm and introduce our motivation. In Section 3.3, we will describe our each component of Prism's framework respectively. In Section 3.4, we illustrate the training and inference process of Prism respectively.

### 3.1 OVERVIEW

Previous methods mainly employ semantic prior to generate 3D assets, which are considered to exhibit less control over the topological structures. To tackle this problem, we propose a novel 3D generation paradigm, namely Prism(a paradigm driven by structural concept prior). Figure 3 shows an overview framework. We propose three core components in Prism: Prior Representation Construction, Shape Encoding, and 3D Space Mapping, with details described below.

### 3.2 3D ASSETS GENERATION PARADIGM DRIVEN BY STRUCTURAL CONCEPT PRIOR

Diffusion models have brought impressive impacts to generative tasks in 2D domain, but they still lag behind in the field of 3D generation. Pre-trained 3D generative models can generate 3D models of a certain quality in appearance. However, these methods can not well control the topological structures of generated 3D assets. This is because during the training process, these methods are trained on the semantic level, without sufficient information related to structural concept.

Different from traditional controllable 3D generation methods which mainly focus on the appearance of generated results, we aim to provide a method for 3D generation that ensures good topological structures while also containing high-quality shape details. So we propose Prism, which adopts structural concept prior to control topological structures and real-world images to control the shape details.

Inspired by Analytic Concept Sun et al. (2024b), which explicitly expresses the topological structures and affordance knowledge of objects by procedure language, we propose multi-modal representations which integrate structural concept prior from Analytic Concept and visual prior from real-world images to form our prior representations, enabling the generated 3D models simultaneously possess good appearance, good topological structures and consistent affordance knowledge with Analytic Concept. Additionally, during the training process, we use the real 3D models as supervision, enabling our model learn the mapping from prior representations to 3D space.

It should be noted that our prior representations hold a one-to-one correspondence with structural concept described in Analytic Concept, ensuring the mapping from the concept space to the 3D space when learning the transfer from prior representations to 3D latents.

Compared with previous 3D generation methods, our generative 3D assets have high-quality appearance, contain rich structural concept information and can align well in affordance knowledge which described in procedure in Analytic Concepts.

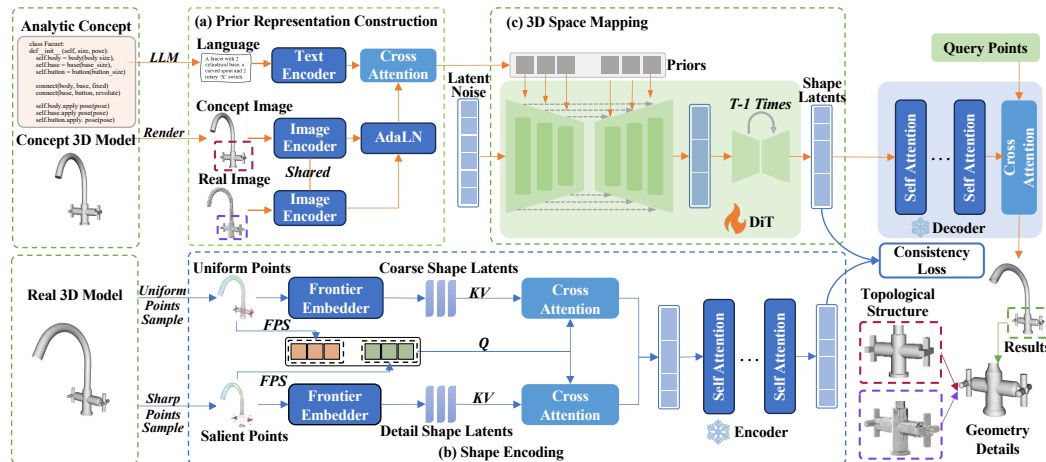

Figure 3: Overall architecture for our method. Blue arrows mean those processes only perform in training stage. (a) In Prior Representation Construction, we learn a novel prior representation which integrated structural concept and shape details by AdaLN Modulation and cross attention. (b) In Shape Encoding, we sample uniform points and salient points from real 3D models and integrate them by cross attention. Then we adopt pre-trained VAE encoder to get the shape latents, serving. (c) In 3D Space Mapping, we fed prior representations into DiT as condition and map it into 3D latent space supervised by shape latents produced in (b). Then we adopt pre-trained VAE decoder to generate 3D assets controllable topological structure and high-quality 3D shape from the latent features produced in DiT.

## 3.3 PRISM FRAMEWORK

### 3.3.1 PRIOR REPRESENTATION CONSTRUCTION

Prior Representation Construction aims to learn better representations that can integrate prior of both structural concept (which can describe the topological structures) and shape details. To achieve this, we employ Analytic Concept to control topological structures and real-world images to control shape details.

Specifically, regarding the structural concept described by Analytic Concept as $C$, we convert $C$ into the corresponding conceptual 3D models $C_m$, and select the appropriate camera parameters to render them into conceptual images denoted as $C_i$. Moreover, we convert the procedures into language descriptions via Large Language Model (LLM) which are represented as $C_t$, thereby obtaining conceptual image-text pairs $\{C_i, C_t\}$. Such multi-modal representations allow for a more comprehensive description of structural concept, enabling the model to learn better prior representations. It should be noted that $C_t, C_m, C_i$, and $C$ are in one-to-one correspondence, each serving as a description of the structural concept.

Besides, we inject real-world images $R_i$ into visual prior to control the shape details of generated 3D assets. The real-world images $R_i$ are first fed into a pre-trained image encoder $\gamma_i$ to obtain corresponding features $F_r$:

$$F_r = \gamma_i(R_i). \tag{1}$$

To well integrate the shape prior with the structural concept prior, we employ an AdaLN (adaptive layer normalization) modulation introduced in Li et al. (2024). We apply this modulation to each attention sub-layer of the pre-trained image encoder $\gamma_i$ when encoding $C_i$, and the modulation layers are optimized during training to better fusion these two types of prior. Therefore, the prior of our visual modality consist of two parts: structured concept and shape details, which can be written as follow:

$$F_v = \gamma_i(C_i, ModLN(F_r)), \tag{2}$$

where $ModLN$ represents the modulation mentioned above. After obtaining the features of the visual modality, we employ a pre-trained text encoder $\gamma_t$ to get the language features:

$$F_t = \gamma_t(C_t), \tag{3}$$

where $C_t$ is the language descriptions of the structural concept. Then we calculate the features $F_m$ of $F_v$ and $F_t$ via cross attention to form our final prior representations:

$$F_m = \text{CrossAttn}(F_t, F_v, F_v). \tag{4}$$

So far, we have obtained the final prior representations $F_m$, which can control not only the structural concept but also the the shape details of generated 3D assets.

### 3.3.2 SHAPE ENCODING

Shape Encoding aims to construct 3D shape latents from real 3D models. Specifically, to better represent the appearance of real 3D models, we employ uniform points sampling to capture the coarse shape and sharp points sampling strategy to capture the geometric details in the form of point clouds. Then we employ the frontier embedder to map these two point clouds into coarse shape latents and detail shape latents respectively. After that, we adopt cross attention to integrate the coarse shape latents and detail shape latents and conduct self attention to obtain final 3D shape latents. The detail process is described in A.4

### 3.3.3 3D SPACE MAPPING

In this section, we employ a learnable DiT Peebles & Xie (2023) to map the prior representations into 3D latent space. We use the 3D shape latents produced in Shape Encoding as supervision and employ consistency loss to align it with the latents produced by DiT, enabling the diffusion model to generate 3D latents that conform to the 3D latent space.

Finally, we leverage the rendering capability of the pre-trained VAE decoder, which decodes the shape latents to generate 3D meshes, thereby achieving the generation of 3D assets from structural concept.

In this way, we can generate high-quality 3D assets. These 3D assets not only have good visual appearance but also possess consistent topological structures along with the affordance knowledge described in Analytic Concept, thanks to the encoded multi-modal prior representations.

### 3.4 TRAINING AND INFERENCE PROCESS

**Training Process.** During the training process, we only train the diffusion model and freeze the pre-trained VAE.

We use the shape latents $G$ obtained in the Shape Encoding process as supervision to train the DiT with fusion features $F_m$ as conditional prior, enabling the latent variables generated by the DiT align with the 3D shape latents. This allows our DiT learn the ability to map from prior representations to the 3D latent space. The conditional DiT is learned in the following way:

$$\mathcal{L}_{\text{DiT}} := \mathbb{E}_{G,y,\epsilon \sim \mathcal{N}(0,1),t} \left[ \| \epsilon - \epsilon_\theta (G_t, t, F_m) \|_2^2 \right], \tag{5}$$

where $\epsilon_\theta$ represents the DiT, which is built on a UNet-like transformer, $t$ is sampled from $\{1, \ldots, T\}$, and $G_t$ is the noisy version of $G_0$.

In addition, we adopt the classifier-free guidance (CFG) Ho & Salimans (2022) training strategy, randomly dropping conditions during training to improve the diversity of generation results.

**Inference Process.** During the inference stage, we only utilize the trained diffusion model and the decoder structure of the pre-trained VAE. The decoder decodes the latent features produced by the diffusion model to predict the occupancy field and then converts it to meshes. Since we froze the pre-trained VAE model during training, the decoder's original rendering capability remains intact.

Moreover, the consistency constraint ensures that the output of DiT is consistent with the 3D latent features output by the encoder. This enables the model to convert our prior presentations into 3D space. The resulting 3D assets can possess good appearance, topological structures, and affordance knowledge.

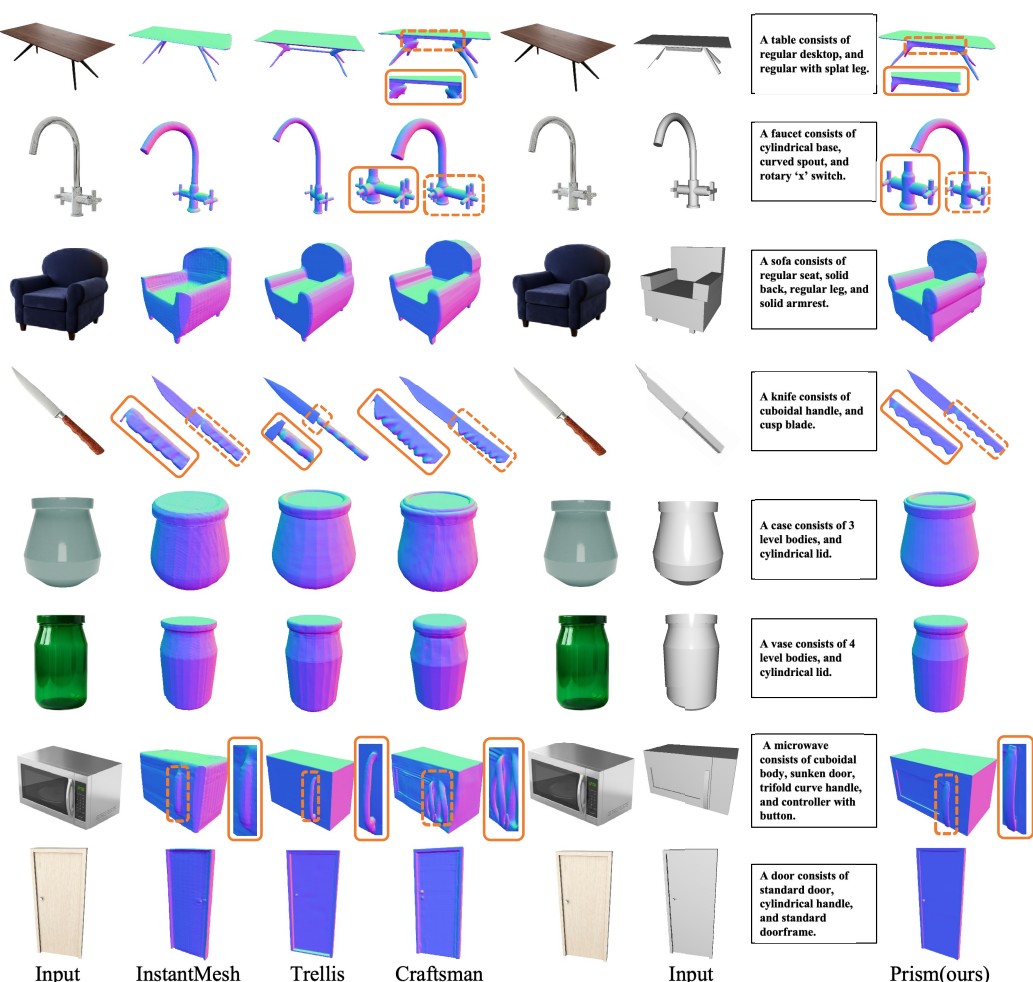

Figure 4: Qualitative comparisons with baseline methods on generation results.

## 4 EXPERIMENTS

### 4.1 IMPLEMENTATION DETAILS

We evaluate Prism on PartNet dataset Mo et al. (2019) with Analytic Concept as annotations, and randomly split the dataset into a training set and a test set at a 3:1 ratio. We adopt Dora-VAE Chen et al. (2025a) as the pre-trained model which contains 1.3 billion parameters. We freeze the VAE model during training. Our DiT has comprising 1.04 billion parameters and conditions on image features extracted by DINOv2 Oquab et al. (2023) and text features extracted by CLIP(ViT-L-14) Radford et al. (2021). Our training is conducted on 4 A100 GPUs for two days with a batch size of 32 and a learning rate of 5e-5. We set CFG with a drop rate of 0.1 and set CFG strength to 3 and sampling steps to 50 at inference stage. We employ Flash-Attention-v2, mixed-precision training with FP16 and gradient checkpointing to optimize memory usage and training efficiency. For manipulation experiments, we adopt SAPIEN Xiang et al. (2020) as simulation environment, using a Franka Panda gripper as the robot actuator. More implementation details can be found in Appendix. A.5.1.

### 4.2 MAIN RESULTS

We evaluate the quality of our generated 3D models from three aspects: appearance, topological structure consistency, and affordance knowledge alignment. Our model is inherently generative

| Method | Concept prior | Semantic prior | FID↓ | CD↓ | Uni3D↑ |
|--------|:---:|:---:|:---:|:---:|:---:|
| InstantMesh | ✗ | ✓ | 23.72 | 0.0316 | 0.20 |
| CraftsMan | ✗ | ✓ | 17.61 | 0.0227 | 0.25 |
| Trellis | ✗ | ✓ | 13.35 | 0.0183 | 0.27 |
| Prism | ✓ | ✓ | **9.15** | **0.0092** | **0.34** |

Table 1: Quantitative comparison with baseline methods on Partnet dataset.

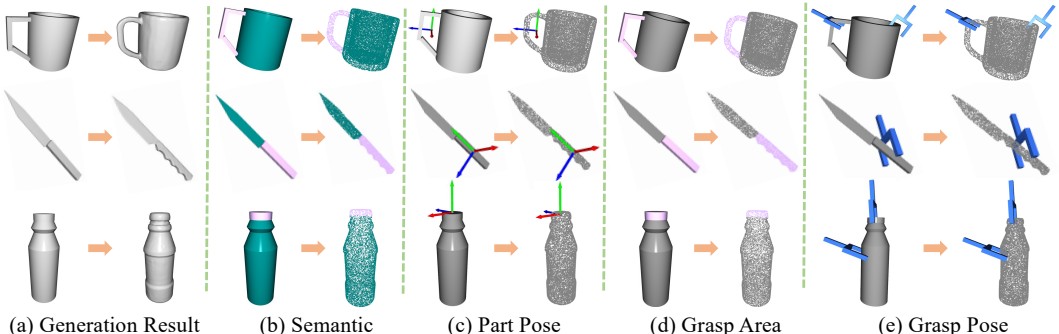

(a) Generation Result    (b) Semantic    (c) Part Pose    (d) Grasp Area    (e) Grasp Pose

Figure 5: Visualization results of generated 3D models with Analytic Concept. Our methods can highly control the topological structures along with part pose, grasp area, and grasp pose described in Analytic Concept.

rather than reconstructive, so it is inappropriate to directly compare our outputs with ground truth in the same way that is standard for reconstruction models. So for appearance quality, we choose Fréchet Inception Distance (FID) and Chamfer Distance (CD) to compare the distributions of our generative 3D models and the real 3D models. Then, we evaluate the topological structure consistency using the Uni3D Zhou et al. (2023) score. For affordance knowledge alignment, we present visualization results on four dimensions (semantic, part pose, grasp area and grasp pose) between Analytic Concept and the generated results.

Our primary focus is on the qualitative quality of the 3D generation results through a variety of results and we also present quantitative data for reference.

### 4.2.1 QUALITATIVE COMPARISON

We compare our method with state-of-the-art baselines including InstantMesh Xu et al. (2024a), CraftsMan Li et al. (2025), and Trellis Xiang et al. (2025). We employ the Partnet Mo et al. (2019) dataset to evaluate the performance of Prism, which is shown in Figure 4. For each category, we choose appropriate camera parameters to better showcase the topological structures and use image with a resolution of 256x256 as input.

Prism exhibits superior performance in controlling topological structures, while also handle shape details well. On the contrary, InstantMesh tends to produce noisy surfaces and incorrect topological structures. Trellis also cannot well control the topological structures, such as connection between the body and handle of knife. CraftsMan also suffers from inaccurate topological structures, such as the position of the switch on the faucet and the layout of legs of the table.

### 4.2.2 QUANTITATIVE COMPARISON

First, we adopt FID and CD to evaluate the appearance of generated results. Then, we use the Uni3D score to calculate the features similarity between the generated 3D models and the real-world image to evaluate the topological structure consistency.

As shown in Table 1, our approach achieves the best performance on all criteria, which illustrates that Prism can maintain topological structures and shape details better than other state-of-the-art baselines.

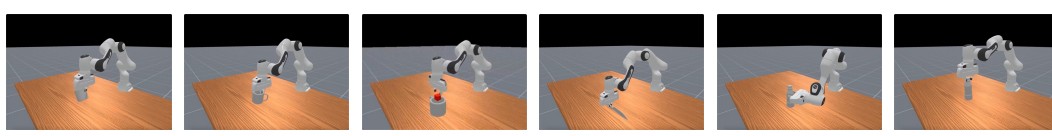

(a) Pick Mug Handle (b) Pick Mug Body (c) Pour Cube into Mug (d) Pick Knife Handle (e) Pick Bottle Body (f) Pick Bottle Lid

Figure 6: Visualization on simulation environment.

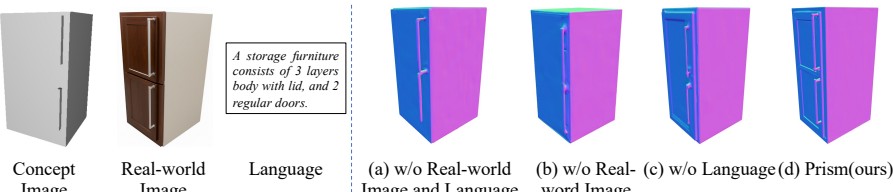

Concept Image Real-world Image Language | (a) w/o Real-world Image and Language (b) w/o Real-word Image (c) w/o Language (d) Prism(ours)

Figure 7: Visualization results of ablation on different components of prior.

### 4.2.3 RESULTS ON MANIPULATION

Since Prism is an object-level generation method, we conduct experiments at the object level. We employ SAPIEN Xiang et al. (2020) as the simulation environment and use a random selected mug, bottle and knife generated by Prism, with Analytic Concept as annotations to evaluate affordance knowledge alignment.

As shown in Figure 5, the 3D models generated by Prism can align well with Analytic Concept across four dimensions: semantic, pose, grasp area and grasp pose. The results indicate that our generation results exhibit good consistency with Analytic Concept prior in terms of structural concept and affordance knowledge. The simulation results are shown in Figure 6, for more details please refer to Appendix A.5.3 and supplementary materials.

### 4.3 ABLATION STUDY

**Ablation on Prior Representation Construction.** In this part, we conduct an ablation study on different components of our prior based on conceptual images which serve as the main carrier of structural concept. The ablation includes scenarios with only conceptual images, without language descriptions, and without real-world images. The quantitative results is shown in Appendix A.5.4.

The visualization results are shown in Figure 7. In Figure 7(a), the generated storage furniture only has the overall structure without shape details in door and handle. And in Figure 7(b), the result cannot fully represent the shape details in the door area. In Figure 7(c), the result can exhibit some shape details but cannot preserve the topological structures. In Figure 7(d), our original prior representations achieve the best performance in both topological structures and shape details.

For more ablation study please refer to Appendix A.5.4.

## 5 CONCLUSION

We present Prism, which leverages structural concept as prior in 3D generation for the first time, aiming to better control the topological structures and shape details, which is crucial in the physical world applications. At its core, the novelties of Prism are as follows: First, we construct novel prior representations which have a one-to-one correspondence with structural concept and learn a mapping between the prior representations and 3D space. Second, to control both topological structures and shape details, we employ AdaLN modulation and cross attention to fuse features from both types of prior, forming the final prior representations. We comprehensively evaluate the performance of Prism from both visual and robotic aspects. The results indicate the superiority of Prism. Despite its ability to generate high-quality 3D meshes in terms of topological structure and shape details aspects, Prism is an object-level generation method. There is still ample room for future exploration, such as part-level generation.

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

# A  APPENDIX

## CONTENTS

## A.1  THE USE OF LARGE LANGUAGE MODELS (LLMS)

In paper writing, we only employ LLMs for polishing.

## A.2  MORE RELATED WORK

### A.2.1  NEUROSYMBOLIC SYNTHESIS

Neurosymbolic Synthesis has long been used to represent graphical content, offering interpretable parameters, stochastic variability, and high-quality outputs, but it is challenging to design from scratch. Advances in AI have enabled neurosymbolic models to combine the strengths of AI and symbolic programs for representing, generating, and manipulating visual data. Transformer-based methods Ganin et al. (2021); Seff et al. (2022); Khan et al. (2024) leverage large-scale sketch datasets for engineering sketch generation, with advances in controllability through image or sketch conditioning, codebook-based geometry/topology separation. Recent research Ellis et al. (2019); Reddy et al. (2021); Xu et al. (2024b) tackles inferring 2D shape programs from images or sketches using reinforcement learning, differentiable rendering or diffusion model to capture structural regularities. Another line of work Xu et al. (2022); Hossain et al. (2025) employs neurosymbolic models to generate 3D shapes by leveraging transformer-based architectures specialized for CAD operations. Although procedural models have achieved promising results across various domains, they remain limited to shape modeling and overlook the conceptual nature of the generated objects. Our work leverages structural concept for 3D asset generation.

## A.3  MORE DETAILS ON ANALYTIC CONCEPT

Analytic Concept represents the structural concept through procedural languages, providing models with a way to perceive, reason, and interact with the physical world. As shown in Figure 8, it explicitly expresses the topological structures and affordance knowledge of objects using mathematical language Sun et al. (2025), thereby modeling objects in the physical world with formalized languages.

In the fields of cognitive science and brain science Biederman (1987), it has been found that humans' understanding of real-world objects stems from the perception of geometric shapes, which is combined with relevant common-sense knowledge for induction. Based on this, Analytic Concept has built a library of geometric primitive templates. By calling different geometric primitive templates in the library, the structural concept information can be described, thus constructing the topological structures of the object itself. Then, the affordance knowledge of geometric templates are defined through procedural languages, including grasping postures, contact points for pushing, and so on.

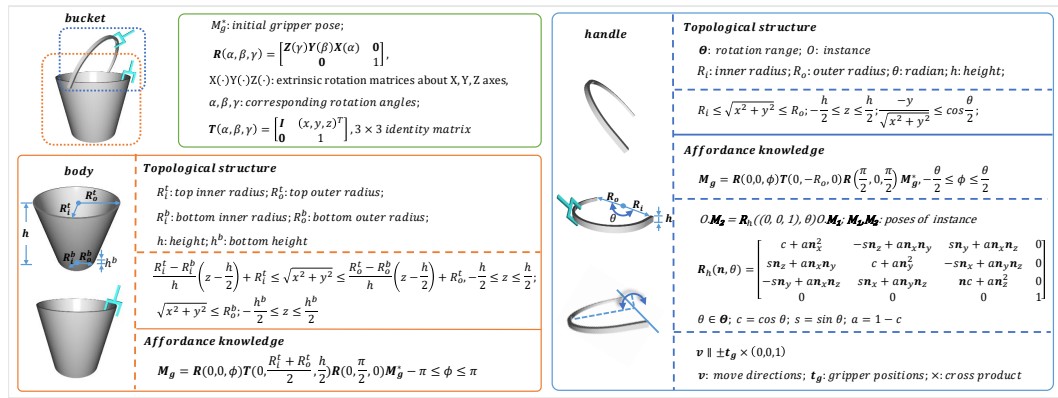

Figure 8: Illustration of Analytic Concept describe in mathematical language, which takes a bucket as example. [**Left**] Describe the component of bucket body, from top-to-bottom is its topological structures definition and affordance knowledge description. [**Right**] Describe the component of bucket handle, from top-to-bottom is its topological structures definition, affordance knowledge description, and kinematic description.

Each geometric primitive template consists of two parts: one is a program to describe geometric properties, and the other is a program to assign corresponding parameters for instantiation. The parameters also include two parts: one is the internal parameters describing geometric properties (such as length, width, height, and diameter), and the other is the external parameters describing the overall shape, including the position and direction.

For each object, constraints are applied to each geometric template, which can be used to represent the connection mode between two geometric templates. At the same time, due to the extremely complex geometric shapes of some objects (for example, a locker with drawers may consist of dozens of geometric shapes), the description program of the object becomes very complicated. To simplify the representations, Analytic Concept establishes advanced primitive templates that bridge the geometric templates and objects. Advanced primitive templates are built based on a set of basic primitives with specific spatial layouts and their connection relationships. For example, the handles of buckets and kettles can be represented by the same advanced primitive template.

The description of an object itself can first call the advanced primitive template to build a subclass and pass corresponding parameters for instantiation Sun et al. (2024a). The advanced primitive template is a subclass inherited from the basic geometric primitive template and instantiated through corresponding parameters, including two parts: continuous parameters describing the shape and pose of the object, and discrete parameters describing the number of repetitions of the geometric primitives.

## A.4 MORE DETAILS ON SHAPE ENCODING.

First, we use a pre-trained VAE model to encode real 3D models, thereby obtaining latent codes for 3D space. To better represent the appearance of real 3D models, we perform uniform points sampling strategy on 3D models to obtain a point cloud $P_s$ that represents the coarse shape of the objects. On this basis, we further detect and sample the sharp edges of each 3D models, so as to obtain a point cloud $P_g$ with sufficiently rich geometric details. We believe that the geometric details of a 3D models can be represented by sampled sharp points from sharp edges. We achieve the sampling of sharp points through two steps:

**Sharp Edge Detection.** Given the mesh of an object, we determine whether an edge is a sharp edge based on the size of the dihedral angle between two adjacent faces, because the size of the dihedral angle can directly reflect the curvature of the mesh edge. A large dihedral angle indicates that the edge is a sharp edge.

For the common edge $e$ of any two adjacent faces $f_1$ and $f_2$. The dihedral angle $\theta_e$ is computed as follows:

$$\theta_e = \arccos\left(\frac{n_{f_1} \cdot n_{f_2}}{\|n_{f_1}\|\|n_{f_2}\|}\right), \tag{6}$$

where $n_{f1}$ and $n_{f2}$ are the normal vectors of $f_1$ and $f_2$ respectively. The set of sharp edges $\mathcal{E}$ includes all edges with a dihedral angle exceeding the preset threshold $\tau$:

$$\mathcal{E} = \{e \mid \theta_e > \tau\}. \tag{7}$$

Let $N_{\mathcal{E}} = |\mathcal{E}a|$ denote the number of sharp edges. The threshold is a hyperparameter set as 10 degrees, and sharp edges are all edges with a dihedral angle exceeding this threshold.

**Sharp Points Sampling.** After obtaining the set of sharp edges, we aim to sample sharp points from the sharp edges to get the point cloud $P_g$ represent the geometric details. Let $P_s$ be the set of sharp points, which is initially an empty set. For each sharp edge, we retain its two vertices $v_1$ and $v_2$ and add them to the set of sharp vertices: $P_r = P_r \cup \{v_1, v_2\}$. After obtaining $P_s$, according to the preset sharp point threshold.

When the number of sharp vertices is sufficient ($N_d \leq N_V$), we use FPS (Farthest Point Sampling) to down-sample $P_{\mathcal{E}}$ to obtain $P_g$. When the number of sharp vertices is insufficient ($N_V < N_d$), we include all vertices in $P_{\mathcal{E}}$ and supplement them with interpolated points $P_I$. These additional points are generated by uniformly sampling $(N_d - N_V)/N_{\mathcal{E}}$ points on each sharp edge in $\mathcal{E}$ to ensure comprehensive coverage of the sharp features.

Therefore, we obtain uniformly sampled point clouds $P_s$ to represent the coarse shape and the sharp sampled point clouds $P_g$ to represent the geometric details. Then we employ fourier feature embedding(FFE) to get the corresponding embeddings:

$$E_s = \text{FFE}(P_s), E_g = \text{FFE}(P_g). \tag{8}$$

We encode $E_s$ and $E_g$ through a dual cross-attention mechanism, enabling the encoder focus on both coarse shape information and fine-grained geometric information. After obtaining dense point cloud representations, we adopt FPS to down-sample $P_s$ and $P_g$:

$$P_f = \text{FPS}(P_s, N_s) \cup \text{FPS}(P_g, N_g), \tag{9}$$

where $N_s$ and $N_g$ are the numbers of point clouds down-sampled from $P_s$ and $P_g$ respectively, resulting in $P_f$. Then, we calculate cross-attention features for uniform points and sharp points respectively:

$$\begin{aligned} G_s &= \text{CrossAttn}(P_f, E_s, E_s), \\ G_g &= \text{CrossAttn}(P_f, E_g, E_g). \end{aligned} \tag{10}$$

Finally, the two attention features are concatenated to obtain the final point cloud features, which serve as the latent space representations of the real 3D geometric information $G = G_s + G_g$. This design can focus on both coarse shape and geometric details respectively, during the feature extraction process. And form a better representation of latent 3D space.

## A.5 MORE EXPERIMENT RESULTS

### A.5.1 MORE IMPLEMENTATION DETAILS

**Data Processing.** Due to significant noise in geometry and appearance, we exclude low-quality meshes from our training data including those with thin structures, holes, and texture-free surfaces, to guarantee high data quality. This process yielded a refined training dataset containing roughly 3k objects and a test dataset with around 1k objects including 16 categories: scissor, mug, chair, Bucket, bottle, table, faucet, knife, refrigerator, display, microwave, trash can, door, vase, dishwasher, and storage furniture.

For mesh processing, we follow CLAY Zhang et al. (2024) mesh to ensure watertight 3D models. For sharp edge sampling, we set the number of sampled points $N_g = N_s = 32768$, and sampling angle threshold $\tau = 10$ degrees.

For each mesh, we construct an image for conceptual representations using procedural language in Analytic Concept. For details, we select an appropriate view to render the concept meshes into

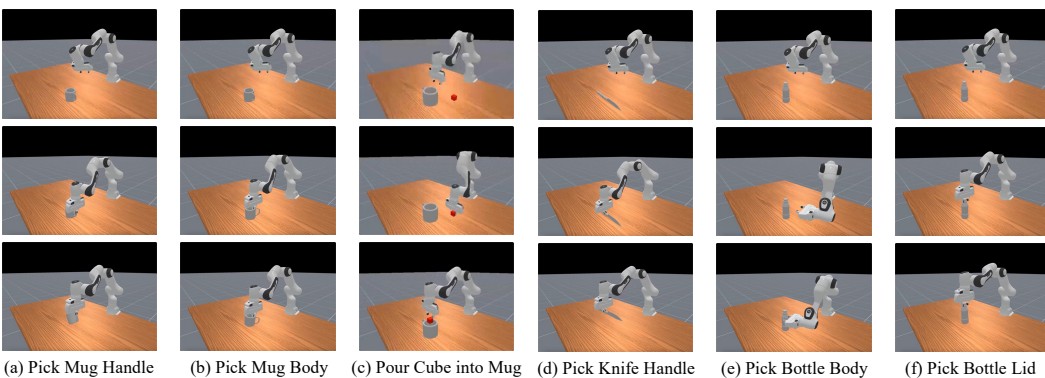

(a) Pick Mug Handle  (b) Pick Mug Body  (c) Pour Cube into Mug  (d) Pick Knife Handle  (e) Pick Bottle Body  (f) Pick Bottle Lid

Figure 9: Results of simulation experiments in SAPIEN simulator.

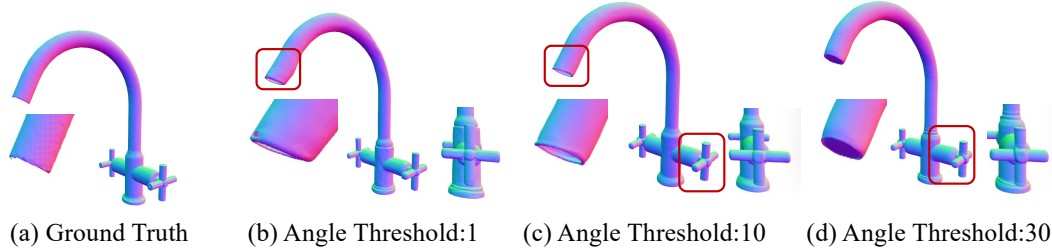

(a) Ground Truth  (b) Angle Threshold:1  (c) Angle Threshold:10  (d) Angle Threshold:30

Figure 10: Visualization results of ablation on different dihedral angle thresholds(/degree).

images which can best represent its topological structure and fuse with real-world images to serve as visual prior. We adopt DALL-E3 Betker et al. to produce real-world images based on real 3D models rendered by the same camera parameters which used to render conceptual images, and transfer procedure language into nature language descriptions to form as the language conditions of prior representations by GPT-4o Hurst et al. (2024).

**Evaluating Metrics.** We conduct a quantitative analysis of the experimental results from two aspects. One is to measure the visual effect of the generative model, which is used to determine the distribution difference between generated 3D model and ground truth 3D model. The other is to measure the consistency of topological structure between the generated 3D model and Analytic Concept, which is used to determine whether the generated 3D model is consistent with the structural concept.

We use FID and CD to measure the appearance between ground truth shapes and generated shapes. We introduce the Uni3D to calculate the feature similarity between real-world images and the generated 3D models. Such a measurement can be used to measure the consistency of topological structures between the two modalities. A higher similarity indicates a higher topological structure consistency.

**Simulation Experiment Setting.** We adopt the SAPIEN Xiang et al. (2020) simulator as the simulation environment for our evaluation. In each manipulation simulation, the target object is initially placed following the annotation position and rotation in Analytic Concept within the simulator. To interact with the target objects, we employ a Franka Panda gripper with two fingers. We consider primitive action of pick up generated 3D models through the corresponding Analytic Concept annotation to evaulate the alignment in affordance knowledge.

### A.5.2 MORE QUALITATIVE COMPARISON RESULTS

We present additional examples of 3D assets generated by Prism and compare with other baselines in Figure 11 to Figure 15.

| Angle threshold | FID↓ | CD↓ | Uni3D↑ |
|---|---|---|---|
| 1 degree | 12.70 | 0.0109 | 0.34 |
| 30 degree | 11.64 | 0.0104 | 0.32 |
| 10 degree | **9.15** | **0.0092** | **0.34** |

| prior Components | | | FID↓ | CD↓ | Uni3D↑ |
|---|---|---|---|---|---|
| $C_i$ | $R_i$ | $C_t$ | | | |
| ✓ | ✗ | ✗ | 18.72 | 0.0174 | - |
| ✓ | ✓ | ✗ | 10.96 | 0.0097 | 0.31 |
| ✓ | ✗ | ✓ | 14.86 | 0.0154 | - |
| ✓ | ✓ | ✓ | **9.15** | **0.0092** | **0.34** |

(a) Ablation on different angle thresholds;     (b) Ablation on different components of prior.

Table 2: Quantitative results on ablation study. $C_i$ represents conceptual images, $R_i$ represents real-world images, and $C_t$ represents language description of Analytic Concept.

As we can see, Prism exhibits the best topological structures and geometric details on these complex categories such as storage furniture, refrigerator, and dishwasher, especially for the handle. Prism integrated structural concept in the prior representations, which leads to better control ability than previous semantic prior.

### A.5.3 More Simulation Results on Manipulation.

For details, we use the Franka Panda gripper to pick the objects according to the grasp pose described in Analytic Concept, including the handle and body of mugs, the handle of knives, and the lid and body of bottles. As shown in Figure 9, we apply 6 tasks to illustrate that our generated meshes can align well with Analytic Concept in affordance knowledge. Figure 9(a) and Figure 9(b) show that the robot can successfully pick the generated mug, Figure 9(c) shows that the robot can pick cube (as a replacement for water) into the mug, which means the mug is aligned with Analytic Concept. Figure 9(d) shows that the robot can successfully pick up the knife by its handle. Figure 9(e) and Figure 9(f) exhibit the interaction to pick up bottle by handle and body respectively.

The results indicate that Prism can generate 3D assets which are well aligned with Analytic Concept in terms of affordance knowledge. The corresponding videos can be found in supplementary materials.

### A.5.4 More Abaltion Study

**Ablation on Shape Encoding.** In this section, we conduct an ablation study on sharp point sampling under different dihedral angle thresholds.

The quantitative result is shown in Table 2a, when the angle threshold is set to 10, the generative model achieves the best performance on FID, CD and Uni3D scores. And when set it to 1, the FID is highest because sharp sample strategy cannot capture geometric details well. And when set it to 30, the FID increases to 11.64, CD decreases to 0.0104 and Uni3D decreases to 0.32, which indicates that the control ability in geometric details and topological structures become lower.

The visualization results are shown in Figure 10. If the angle threshold is too small (set to 1 degree), the sharp points sampling is difficult to capture the model's sharp edges as the geometric detail for the Shape Encoding module. Conversely, if the angle threshold is too large(set to 30 degree), the sharp points sampling strategy cannot capture enough geometric details, which results in insufficient accuracy of the geometric details generated at the joints and knobs.

**Quantitative Results of Ablation on Prior Representation Construction.** The quantitative results are shown in Table 2b We evaluate Uni3D with real-world images and generated 3D models, so the metric is not available when w/o real-world images. As we can see, when missing real-world image, the FID is higher because the model cannot well control the shape details. When w/o language descriptions, the Uni3D decreases from 0.32 to 0.30, which indicates that the topological structures control ability of such scenario is lower than our original prior representations.

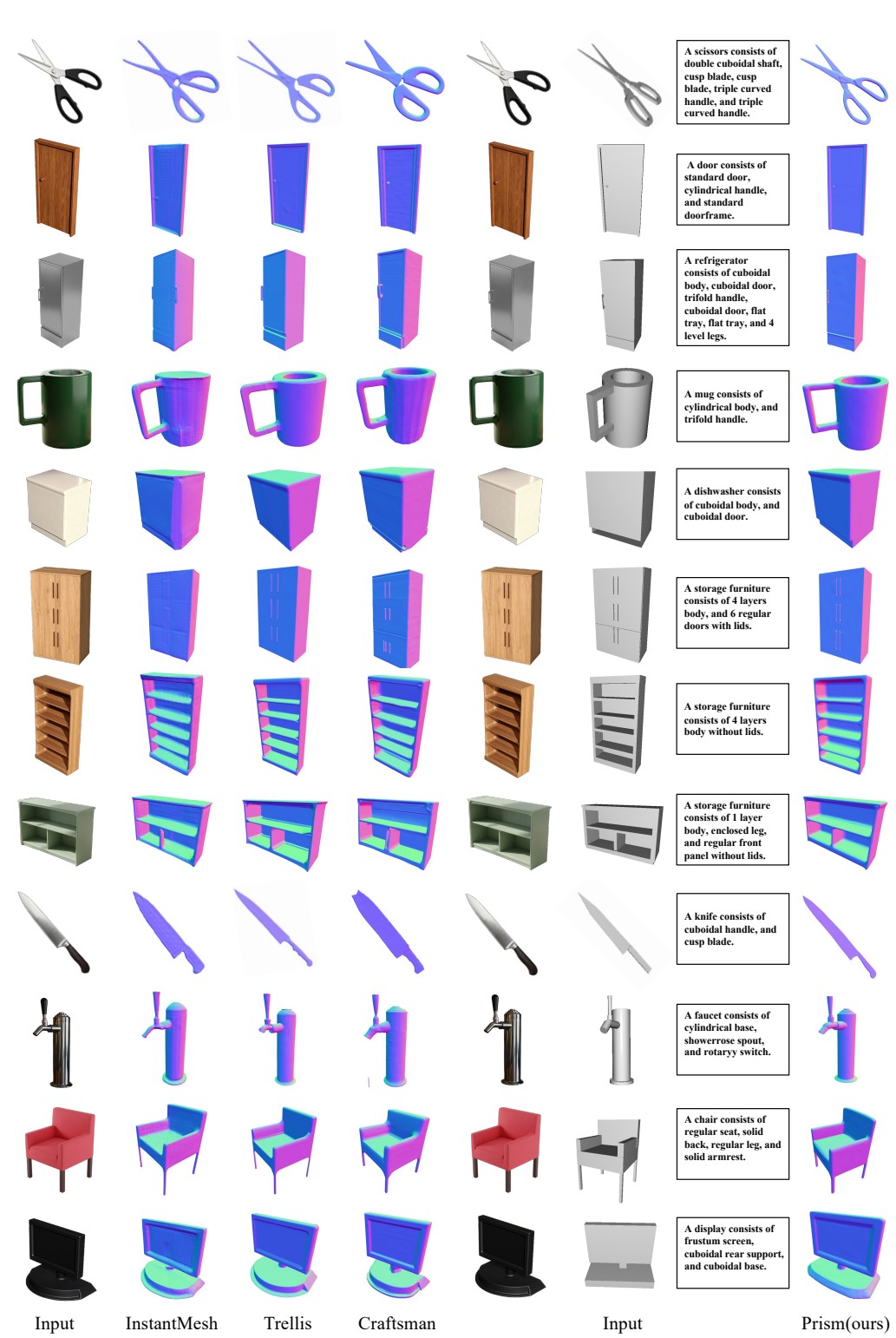

Figure 11: Qualitative comparisons with baseline methods on generation results.

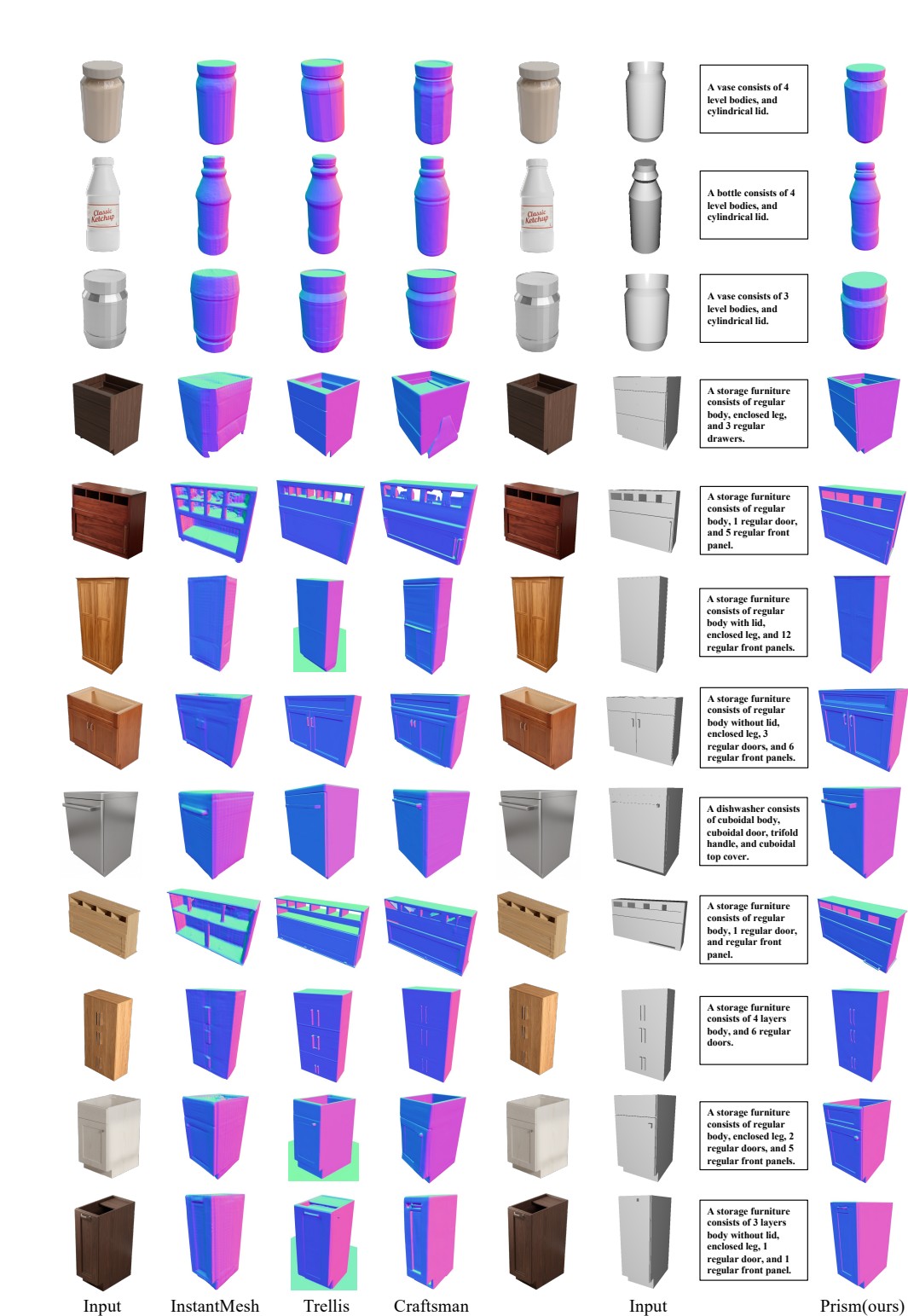

Figure 12: Qualitative comparisons with baseline methods on generation results.

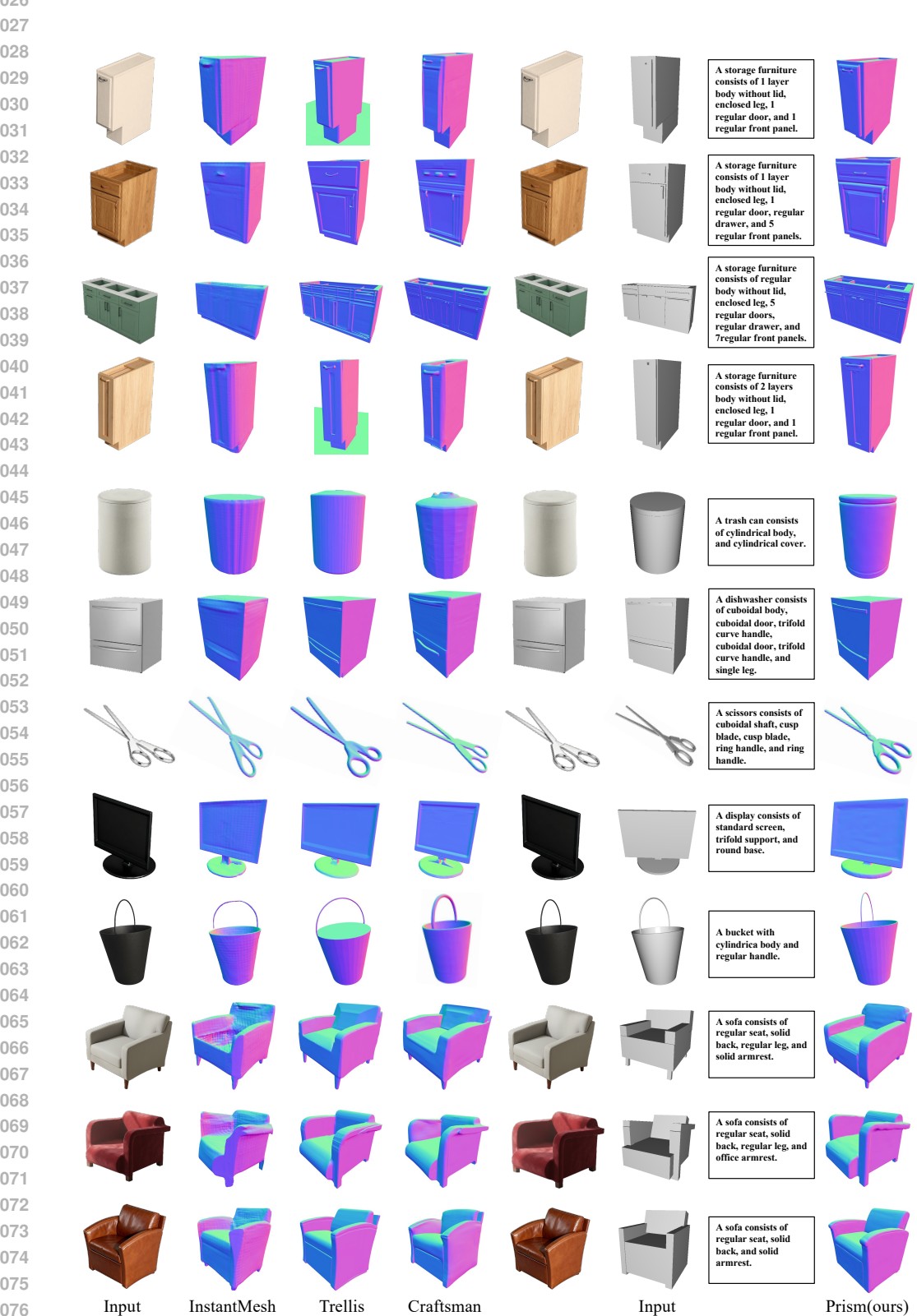

Figure 13: Qualitative comparisons with baseline methods on generation results.

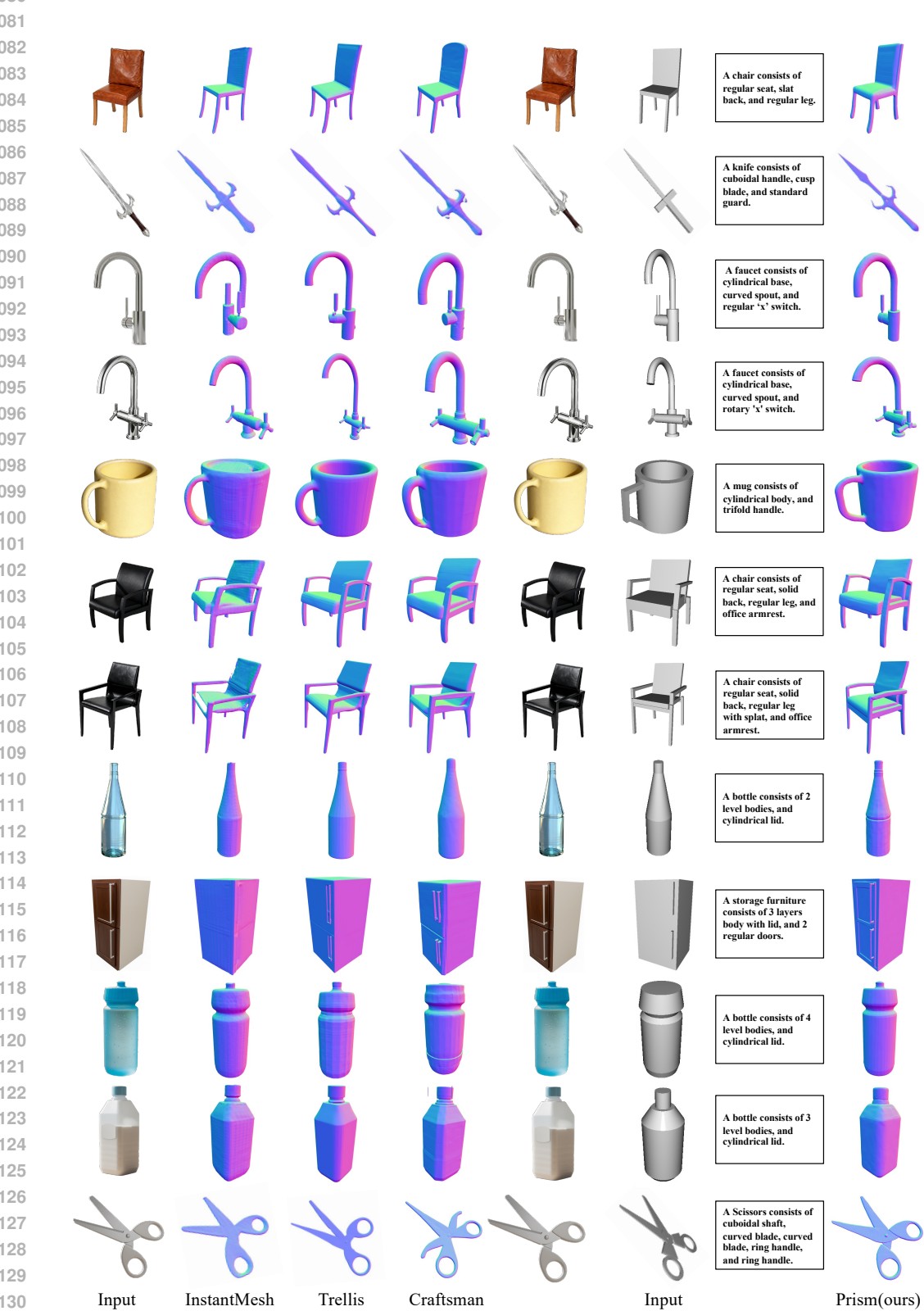

Figure 14: Qualitative comparisons with baseline methods on generation results.

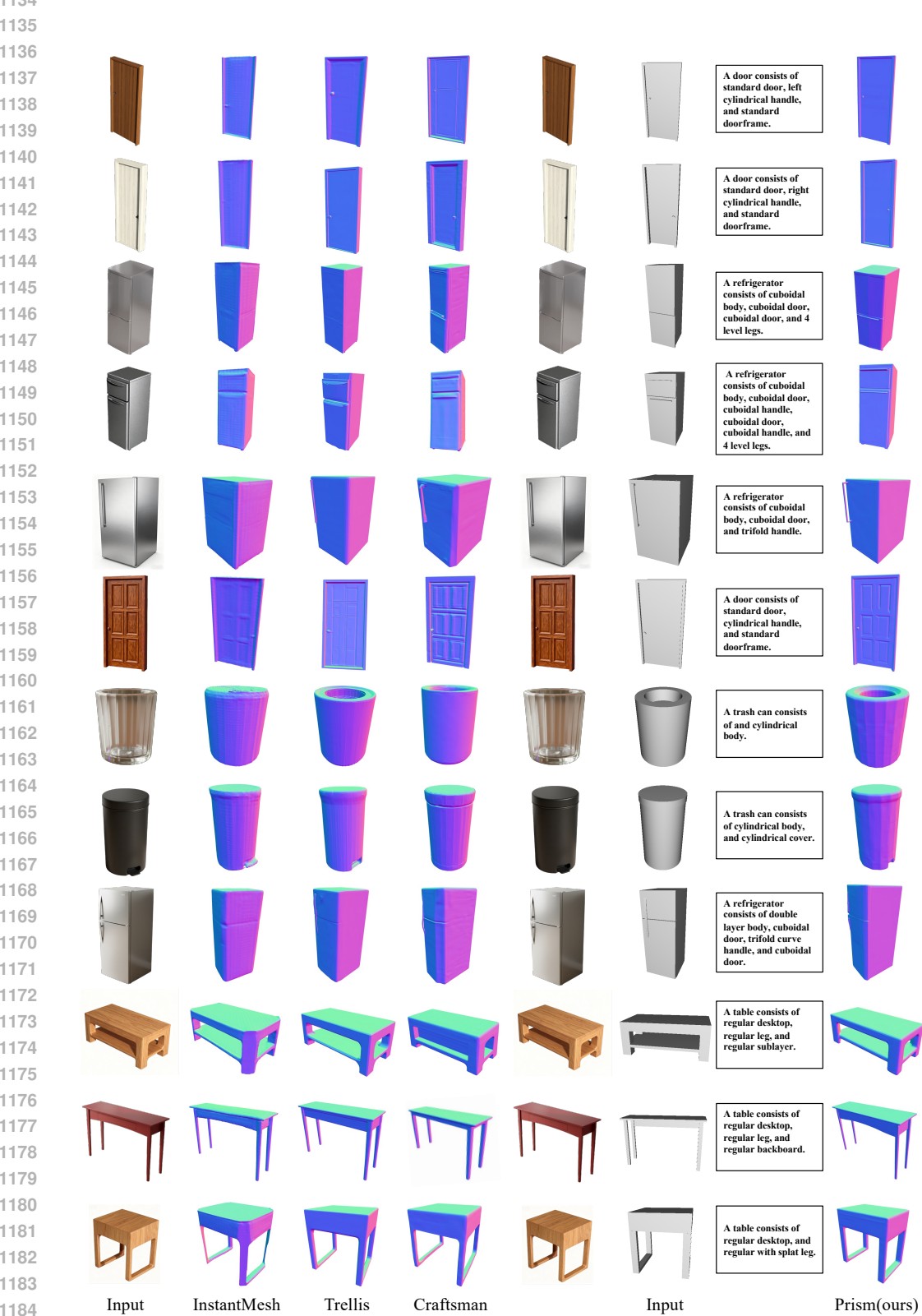

Figure 15: Qualitative comparisons with baseline methods on generation results.

