# OpenReview forum: "PRISM: A Paradigm for Controllable 3D Generation Driven by Structural Concept Prior"
_ICLR.cc/2026/Conference — ICLR 2026 Conference Withdrawn Submission_

### Official Review · Reviewer_QicA · 2025-10-14

**Soundness:** 3
**Presentation:** 2
**Contribution:** 2
**Rating:** 2
**Confidence:** 5

**Summary:**

PRISM: A Paradigm for Controllable 3D Generation Driven by Structural Concept Prior introduces a novel 3D generation framework that prioritizes structural concept prior over traditional semantic priors. The authors argue that existing methods—which rely on 2D diffusion lifting, multi-view image regression, or native 3D distribution modeling—excel in visual realism but fail to offer precise control over an object's topological structure and affordance (i.e., how it can be interacted with). To address this, PRISM leverages Analytic Concept, a procedural language that describes objects in terms of their geometric primitives, topological relationships, and functional affordances.

**Strengths:**

1. Introduction of a Novel and Meaningful Prior: The "Structural Concept"

The most significant contribution of PRISM is its conceptual shift from purely appearance-driven priors to a structural and functional prior.

Beyond Semantics: Existing priors answer "what is this?" (a chair) or "what does it look like?". The structural concept prior answers "how is this built?" and "how is it used?". It encodes topological constraints (e.g., a chair must have a back support connected to a seat) and affordance knowledge (e.g., a mug handle is graspable). The paper operationalizes this abstract prior through Analytic Concept (Sun et al.), which uses a procedural, mathematical language to define objects. For example, a bucket is not just a cylinder; it is defined as a cylindrical_body connected to a regular_handle with specific spatial constraints. This provides a formal, computable representation of structure.

2. Effective Fusion of Heterogeneous Priors for Multi-Faceted Control

Dual Control Mechanism: The framework is designed to control two distinct aspects simultaneously:Topological Structure: Governed by the Analytic Concept (via conceptual images C_i and text descriptions C_t). Shape Details & Realism: Governed by real-world images R_i, which provide rich, realistic textural and stylistic information. This multi-modal fusion is the key to why PRISM works. The ablation study in Figure 7 shows that removing either the real-world image (resulting in a bland, overly schematic output) or the structural concept (resulting in a structurally incorrect output) leads to failure. Only their combination produces high-fidelity, structurally correct assets.

**Weaknesses:**

1. Limited Scope and Scalability: A Method for Simple, Structured Objects

The most significant limitation of PRISM is its confinement to a domain of relatively simple, parametric, and part-based objects, which raises questions about its generalizability.

Narrow Dataset (PartNet): The model is trained and evaluated exclusively on the PartNet dataset, which contains categories like chair, table, mug, and trash can. These objects are inherently decomposable into a small set of geometric primitives (cuboids, cylinders) and are well-suited for description via Analytic Concept.

Struggle with Organic and Complex Geometry: The paradigm appears ill-suited for a vast array of real-world objects that lack a clear, hierarchical part structure. How would PRISM handle:

Organic Forms: A human face, a sculpted statue, a tree, or a crumpled piece of paper.

Complex Topology: A intricate piece of jewelry, a bike chain, or a tangled set of cables.

Non-Rigid Objects: A cloth, a plush toy, or a liquid.
The Analytic Concept library would be incredibly difficult, if not impossible, to define for such categories. The paper's approach is fundamentally tied to a procedural modeling mindset, which breaks down for highly complex or organic shapes.

2. Insufficient Demonstration on Complex Categories

The experiments, while showing clear improvements on simple objects, do not convincingly prove that the method handles "complex" cases within its own domain.

"Complex" is Relative: The paper claims to use a "complex test dataset," but this is self-collected and still consists of the same PartNet categories (e.g., storage furniture, refrigerators). A truly complex example for this method would be a multi-drawer cabinet with intricate handles, a multi-functional tool, or a complex mechanical assembly.

Lack of Testing on complex objects: The qualitative results (e.g., Figures 4, 11-15) primarily showcase success on objects with low part-count. There is no demonstration of its performance on the most structurally dense objects within PartNet (e.g., a detailed lamp with many arms or a complex workstation). The failure cases of PRISM itself are not shown, making it difficult to assess its true boundaries.

Comparison is Unfair to More General Models: PRISM is compared against general-purpose 3D generators (InstantMesh, TRELLIS) that are not designed for structural control. A more telling comparison would be against other structure-aware or CAD-generation methods (e.g., those in neurosymbolic synthesis). Dominating general models on a structured-object-specific task is expected, not revolutionary.

3. Heavy Reliance on a Niche and Costly Annotation System

The entire paradigm is dependent on the existence of Analytic Concept annotations.

Bottleneck for Scalability: Creating Analytic Concept descriptions for a 3D model is a highly specialized and labor-intensive task, requiring expert knowledge to define the procedural rules, primitives, and constraints. This severely limits the potential to scale the training set to internet-scale datasets like Objaverse.

The "Automation" Claim is Weak: The paper mentions using an LLM (GPT-4o) to convert procedures into language descriptions

**Questions:**

1. Limited Scope and Scalability: A Method for Simple, Structured Objects
2. Insufficient Demonstration on Complex Categories
3. Heavy Reliance on a Niche and Costly Annotation System

---

### Official Review · Reviewer_22LA · 2025-10-26

**Soundness:** 1
**Presentation:** 1
**Contribution:** 1
**Rating:** 2
**Confidence:** 4

**Summary:**

This paper claims to present a new paradigm for controllable 3d generation driven by Analytic Concept prior. The Analytic Concept scripts are encoded together with their corresponding real-world and rendering images to condition a 3D DiT generation process. Experiments on PartNet shows better metrics in terms of reconstruction and generation, in comparison with image-to-3D generation baseline methods like Trellis, InstantMesh and Craftsman.

**Strengths:**

N/A

**Weaknesses:**

- Overall, I think this paper is poorly written, see typos below. And there exist many vague and imprecise descriptions. For example, "a novel prior representation ... one-to-one correspondence" as mentioned in the last paragraph of Section 1 is quite confusing. First of all, what is a "prior representation"? Is it the training pairs mentioned in Line 256 or is it the DiT? Moreover, the training pairs are not one-to-one corresponded since it is an ill-posed monocular image reconstruction task.
- The claimed "topological structure" ability is not proved anywhere in the experiments. The metrics FID, CD, and Uni3D are general reconstruction/generation metrics. Moreover, I do not see much topological error for trellis results. These all weaken the contribution and importance of the proposed method.
- The claimed affordance knowledge is brought by the Analytic Concept scripts, not something embedded in the 3d generation process.
- The applicability of the proposed method is questionable, since it requires very complex Analytic Concept scripts and their renderings as input. These are generally not accessible to real world use cases.
- The experiment scale is too small, only on PartNet. There could be serious overfitting issues considering the similarity inside each category of PartNet. And are baselines also trained on the PartNet training set or authors just inference their pretrained weights?
- Only rigid objects can be generated. The articulation is not explored, for example, the microwave in Figure 2 has a door that can be manipulated. But the proposed method fails to consider this. In Figure 4, it seems like the door of the microwave is not even separable from the main body. How is this considered as topologically correct, considering the fact that very detailed Analytic Concept scripts are used as inputs?

Typos:
- Line 30: will be public"ly" available
- Line 124: duplicate "in procedures"
- Line 124: "And can render ..." no subject in this sentence
- Misuse of \citet and \citep, e.g., line 204, 291, section 4.1, and many more...
- Line 368, 5e-5 => 5\times{}10^{-5}

**Questions:**

N/A

---

### Official Review · Reviewer_1Bqb · 2025-10-31

**Soundness:** 3
**Presentation:** 2
**Contribution:** 3
**Rating:** 4
**Confidence:** 3

**Summary:**

PRISM introduces a new paradigm for controllable 3D generation that uses structural concept priors—derived from Analytic Concept—to generate 3D assets that are both visually realistic and structurally coherent. Unlike existing 3D generation methods that rely on semantic or appearance priors, PRISM emphasizes topological structure control and affordance knowledge, aiming to make generated objects more interpretable and functionally grounded for downstream applications such as robotics or simulation. The proposed framework integrates three main components: Prior Representation Construction, Shape Encoding, and 3D Space Mapping.

**Strengths:**

1. This paper introduces a formalized structural concept prior derived from Analytic Concept is innovative. It extends the representation of 3D generation beyond appearance or semantics to include procedural topology and affordance reasoning. The usage of a diffusion transformer for latent alignment to learn a one-to-one mapping from symbolic concept space to geometric 3D latent space is also interesting.
2. The paper validates both visual realism and physical interpretability (robotic manipulation in simulation), covering both perceptual and functional dimensions.
3. PRISM achieves SOTA quantitative scores and visibly improved topology consistency compared to InstantMesh, CraftsMan, and Trellis.

**Weaknesses:**

1. The author should use \citep{} rather than \cite{} in many places, for example in “Neural Radiance Fields (NeRF) Mildenhall et al. (2021)”. The incorrect citation format leads to inconsistent and unprofessional typesetting.
2. The experiments focus on PartNet and small-scale manipulation scenes. How about the generalization to more complex 3D objects, such as the teaser objects in Trellis?
3. Given that Analytic Concept annotations are handcrafted, how scalable is PRISM to diverse object categories lacking such procedural definitions?
4. Can PRISM-generated models be directly integrated into embodied policy learning frameworks (e.g., manipulation or grasp planning) beyond visualization-level simulation?

**Questions:**

See weakness

---

### Note · Authors · 2025-11-14

I have read and agree with the venue's withdrawal policy on behalf of myself and my co-authors.